# A Simple but Effective Combination of pH Indicators for Plant Tissue Culture

**DOI:** 10.3390/plants12040740

**Published:** 2023-02-07

**Authors:** Bryn Funnekotter, Ricardo L. Mancera, Eric Bunn

**Affiliations:** 1Kings Park Science, Department of Biodiversity, Conservation and Attractions, Kings Park, Perth, WA 6005, Australia; 2Curtin Medical School, Curtin Health Innovation Research Institute, Curtin University, P.O. Box U1985, Perth, WA 6845, Australia

**Keywords:** bromocresol green, bromocresol purple, chlorophenol red, cell culture, in vitro culture

## Abstract

The use of pH indicators provides a simple, semi-quantitative visual method for quickly assessing pH changes in tissue culture media; however, pH indicators are rarely used in routine plant tissue culture media. In this study, chlorophenol red, bromocresol purple, and bromocresol green were tested to assess their functionality in the growth medium for plant shoot cultures. In addition, a combination of bromocresol green and bromocresol purple was tested to determine if they would widen the observable colour change to better assess pH changes in the medium. Varying the ratio of bromocresol green to bromocresol purple alters the pH at which the colour changes from blue to green to yellow, with a 1:3 ratio providing a useful pH range of 5–6.5, while a 1:1 ratio provides a useful pH range of 4.5–6. All the pH indicators showed no toxic side effects for the plant species tested in this study and were able to be autoclaved to ensure media sterility. The addition of these pH indicators to quickly assess media pH in large tissue culture collections can aid in routine maintenance. These pH indicators can be used as a ‘traffic light’ system, with blue indicating a high pH, green a normal pH, and yellow a low pH in the media.

## 1. Introduction

The pH of plant tissue culture media is an important factor affecting nutrient availability and the growth of cultures [1,2]. While many plant species in tissue culture can tolerate a wide range of pH values in the range 4.0–7.2, the best growth results are normally obtained with a slightly acidic media, typically around pH 5.8 [1]. Furthermore, the pH of the media is indeed known to affect the morphological characteristics of the explant material, such as the morphogenesis of cell cultures into callus, root, and adventitious shoot cultures [1,3].

The pH of plant tissue culture media is affected by the basal salts and buffering agents added, the autoclaving process, as well as the release of secondary metabolites and organic acids by the plants, all of which can affect the pH over time [4]. The use of pH indicators or halochromic substances provides a simple visual method for quickly assessing pH changes in the tissue culture media, and this has long been used in mammalian cell culture media with the addition of phenol red to indicate when the media requires changing or if there is some possible bacterial contamination of the media [5]. Despite this widespread application, pH indicators are rarely used in routine plant tissue culture media. Bromocresol purple, a tryphenylmethane dye, has been the most commonly used pH indicator in plant tissue culture media, particularly for protoplast culture media, which was first used by Roscoe and Bell [6] for *Petunia*. Bromocresol purple has since been used to assess pH changes in flax regeneration [7,8], protoplast cultures of *Nicotiana* and *Petunia* [9,10], rhizosphere pH around *Vigna unguiclata* roots [11], and banana culture [12]. Bromocresol purple changes colour from violet at pH above 6.8 to yellow when the pH drops to 5.2, providing a useful colour change for the typical pH of plant culture media; however, as discussed above, plants can tolerate lower pH conditions without detrimental effects. Chlorophenol red is another potential pH indicator for plant tissue culture that covers a similar pH range (with a colour change in the pH range 4.8–6.8) and which has been used previously by Kramer et al. [13] for *Zea mays* cultures. For a lower pH indicator, bromocresol green has a similar colour change to bromocresol purple but occurs over a pH range of 3.8–5.4 and has been used to assess pH changes in root cultures of tea [14] as well as Arabidopsis and tobacco cultures [15].

In this study, chlorophenol red, bromocresol green, and bromocresol purple were tested to assess their functionality as a semi-quantitative indicator of pH in the growth media of plant shoot cultures. In addition, a combination of bromocresol green and bromocresol purple was tested to determine if they would widen the observable colour change upon pH change, such that they could be more applicable for the standard pH range that plants can tolerate in tissue culture. The aim of this study was to determine if the pH indicators could be used in a similar capacity as phenol red in animal cell culture media, visually informing the users if the pH of the media has changed and may require subculturing. These pH indicators were tested on ten Australian species maintained in the Kings Park Science tissue culture laboratory for conservation purposes [16]. Little is known about the optimal tissue culture growth conditions for these species, including the effects of media pH, and many would benefit from additional work to optimise the tissue culture process [17].

## 2. Results and Discussion

### 2.1. Colour Change of the pH Indicators

Based on previous reports of pH indicators used in plant cultures, chlorophenol red and bromocresol purple were identified as likely candidates due to their colour changes around pH 6 (Figure 1). However, below a pH of 5.5, both pH indicators showed little colour change, with the most visible colour change occurring in the pH range of 5.5–6.5. Bromocresol green showed a similar colour change to bromocresol purple; however, the change from blue to yellow occurred in the pH range 4–5. Thus, by combining bromocresol green and bromocresol purple, the pH at which the colour change occurs can be altered to better suit the pH range of interest for plants in tissue culture, with the 1:3 (*w/w*) bromocresol green:bromocresol purple ratio providing a useful pH range of 5–6.5, whereas the 1:1 ratio provides a useful pH range of 4.5–6 (Figure 1).

### 2.2. Growth of Cultures on Medium Containing pH Indicators

Initial assessment of chlorophenol red and bromocresol purple was done at concentrations of 5 and 20 µg mL^−1^ to assess a visible colour change and any potential negative side effects on plant growth. Both pH indicators at both concentrations exhibited an easily visible colour change over the 5 week testing period (Figure 2), although the change was more noticeable at the higher 20 µg mL^−1^ concentration, whilst in the lower 5 µg mL^−1^ concentration the colour started fading towards the end of the 5 week period. This trial showed both chlorophenol red and bromocresol purple to be limiting in the colour range for shoot cultures, where the colour of the medium quickly changed to yellow as the pH fell below 5.5; these two pH indicators are most commonly used for protoplast and cell cultures, where the media pH is more tightly controlled around pH 6.0 [6,10,13]. The *C. galeatum* shoots growing on 20 µg mL^−1^ chlorophenol red did show a greater extent of browning on the lower leaves; however, this observation is commonly seen with this species in our laboratory but may indicate the shoots are potentially under some additional stress and higher pH indicator concentrations may become detrimental for this species. Additional tests with *P. basistyla* showed no visual differences in shoot appearance between the pH indicators at either a 5 or 20 µg mL^−1^ concentration (data not shown). Further experiments with the pH indicators were performed at a 10 µg mL^−1^ concentration, at which the medium is clearly dyed while reducing any potential negative side effects from adding these dyes at higher concentrations. This is in a similar range to what these pH indicators have previously been used at, typically at 8 µg mL^−1^ [6,7,9,10], although bromocresol purple has been used at a much higher concentration up to 100 µg mL^−1^ [11].

A large trial of ten Australian plant species was done to test the combination of bromocresol green and bromocresol purple (1:1 and 1:3 ratios) over an eight week period. All species were placed on the same basal medium, starting at approximately pH 5.75. It was observed that the pH dropped by 0.25 pH units after autoclaving (from the pH 6.0 it was set at prior to autoclaving), which is a well-known occurrence where the media pH will change after the autoclaving process [18,19].

Pictures were taken weekly to demonstrate the colour change seen when using the combination pH indicators (Figure 3 and Figure 4). In the 1:3 combination, it was easy to visualise colour changes in the first three weeks before the pH of the medium fell towards 5 for many species (Figure 3). By contrast, the 1:1 combination still exhibited a visual colour change when the medium pH was around 5 (Figure 4). While the 1:3 combination was not as sensitive to pH changes occurring in the first two weeks, it was more informative with a few species where acidification of the medium rapidly decreased to pH 5 or lower (i.e., *A. viridis*, *G. scapigera*, and *P. basistyla*). It was observed that the pH-related colour faded in some of the culture medium towards the end of the eight week period, perhaps as a consequence of possible inactivation or interference with the indicators by secondary metabolites released by some of the species. A higher starting concentration of the pH indicators may help with this colour fading, as seen in the 20 µg mL^−1^ trial (Figure 2). After the eight week period, plant material was removed from the culture vessels, and the medium pH was tested again using the TPS pH Cube with an Ionode IJ44C pH electrode. Most visual estimations of the pH were similar to the final pH value measured by the pH electrode and showed no significant difference in their mean value using a Welch two-sample *t*-test for either the 1:3 combination (*t* = −0.1156, df = 19.417, and *p*-value = 0.9092) or the 1:1 combination (*t* = −0.10424, df = 19.958, and *p*-value = 0.918). The 1:1 combination showed a maximum difference of 0.25 pH units compared to the pH probe results, with a standard deviation of 0.165 across the species tested, while the 1:3 combination showed similar results with a maximum difference of 0.3 pH units (and a 0.156 standard deviation).

The changes to the medium pH seen varied widely between species, with four main responses observed (Figure 5). Two species (*H. rutilans* and *S. bancroftii*) showed very little change in pH over time, with the medium retaining a stable pH around 5.5 from week 1 to week 8. It is interesting to note that this final pH is higher than the control medium at week 8 for these two species (pH 5.5 and pH 5.1, respectively), indicating a possible modifying influence on the medium pH by these species, which is quite different from the other species tested and requires further investigation to determine a cause. It is interesting to note that both these species are able to be maintained on basal medium for extended periods (6–12 months) as part of their normal subculture routine with no detrimental effects, although further work will be needed to see if the stability of the medium pH is linked to these long subculture routines. *Commersonia erythrogyna*, *C. galeatum*, and *E. virens* showed a slow decline in pH over the eight week period, with the medium pH stabilising around pH 5. This decline to pH 5 may slow the growth of these species, but it is unlikely to cause damage to the shoots as many species have been reported to grow in media at this pH, including *Chlamydomonas* [20], *Solanum* [21], and *Ptilotus* [19], but once the media drops below pH 5, the growth of the plants can slow significantly [20], and this may be a good indication point for when to subculture to fresh media.

Three species (*A. viridis*, *G. scapigera*, and *P. basistyla*) showed a rapid decrease in pH in the first three weeks, coinciding in the case of *A. viridis* with shoots starting to produce roots. The large decrease in pH did not seem to affect these three species, all of which showed healthy shoots at the end of the eight week period, although *P. basistyla* did start to exhibit a small amount of yellowing in the leaves and is usually subcultured after 4 weeks to avoid this. It would be interesting to assess if nitrogen uptake by these plants affects the media pH [21], especially along the rhizosphere in *A. viridis*, which showed similar results to chickpea seedlings supplied with nitrate as the nitrogen source, decreasing the rhizosphere pH to 4.5 [22]. The two *Eucalyptus* species (*E. argutifolia* and *E. impensa*) showed an initial increase in pH to approximately 6 in the first week, similar to reports on *Hemerocallis* shoots in culture [23], before the medium slowly acidified to a pH below 5. Both these *Eucalyptus* species require subculturing every four weeks to maintain healthy cultures, at which point we observed the pH of the medium falling below pH 5.5, with *E. argutifolia* starting to show substantial leaf browning by the end of the eight week period. Further work will need to be done to fully understand if the decline in *E. argutifolia* shoot health was related to the acidic medium or if the shoots have depleted the nutrients in the medium.

Overall, both the 1:1 and 1:3 combinations of bromocresol green and bromocresol purple can provide a quick visual assessment of media pH changes over time for all the species tested. A semi-quantitative pH value can be determined based on the colour of the media by comparing it to some pH standards set over the pH range of interest and is accurate to approximately 0.25 pH units in this study (Figure 3 and Figure 4). The addition of these pH indicators highlighted some species that may be affected by changing pH in the media for future work, such as the *Eucalyptus* species (Figure 5). Further testing of the media with a pH probe on these species and testing a variety of basal salts and buffering agents will provide significantly more accurate results for quantitative statistical analysis to assess the role media pH may have in optimising conditions for healthy shoot cultures.

## 3. Materials and Methods

### 3.1. pH Indicators

Three pH indicators were tested in the tissue culture medium: chlorophenol red (BDH Ltd., Poole, England, #755008), bromocresol green (Sigma-Aldrich, St. Louis, MI, USA, #114359), and bromocresol purple (BDH Ltd., Poole, England, #2134000). A 50 mg mL^−1^ stock solution was made for each pH indicator, dissolved in ethanol (Rowe Scientific, Perth, Australia, #CE1738), and stored in amber vials at 4 °C.

### 3.2. Plant Material

All plant tissue culture material was sourced from previously established cultures at Kings Park Science, Department of Biodiversity, Conservation, and Attractions (DBCA). The addition of a pH indicator to the media was tested on ten Australian species: *Anigozanthos viridis* Endl. (Haemodoraceae)*, Commersonia erythrogyna* C.F.Wilkins (Malvaceae)*, Conospermum galeatum* E.M.Benn. (Proteaceae)*, Eucalyptus argutifolia* Grayling and Brooker (Myrtaceae)*, Eucalyptus impensa* Brooker and Hopper (Myrtaceae)*, Eremophila virens* C.A.Gardner (Scrophulariaceae)*, Grevillea scapigera* A.S.George (Proteaceae)*, Hemiandra rutilans* O.H.Sarg. (Lamiaceae)*, Philotheca basistyla* Mollemans (Rutaceae), and *Symonanthus bancroftii* (F.Muell.) Haegi (Solanaceae). These species were selected for this trial as it is not known what effects they may have on the media’s pH or if the media’s pH is linked to culture health. There is a wide range of subculture times varying from a standard 4 weeks (*C. galeatum*, *E. argutifolia*, *E. impensa*, and *P. basistyla*), 8 weeks (*A. viridis*, *E. virens*, and *G. scapigera*) or up to 6 months (*C. erythrogyna*, *H. rutilans*, and *S. bancroftii*), understanding if pH is linked to these varying subculture times would be valuable in optimising the subculture process for these threatened species and any new species that will be initiated into tissue culture.

All species were micropropagated onto a half-strength Murashige and Skoog basal medium as previously described [24], with macro- and micronutrients modified to include a total of 100 μM NaFeEDTA, 1 μM thiamine hydrochloride, 2.5 μM pyridoxine, 4 μM nicotinic acid, 500 μM myo-inositol, 500 μM 4-morpholineethanesulfonic acid (MES), 20.5 g L^−1^ sucrose, 8 g L^−1^ agar, and 0.1 µM 6-benzyladenine. The pH indicators were added at a range of concentrations, from 5 to 20 µg mL^−1^, with the pH set to 6.0 using a TPS pH Cube with an Ionode IJ44C pH electrode, and 40 mL of medium was added into a 120 mL polycarbonate jar and capped with a polypropylene vented lid (0.45 mm micropore vent spot covering a 4 mm diameter vent hole) prior to autoclaving at 121 °C for 15 min.

### 3.3. Experimental Design

The testing of the pH indicators and their combinations was done as a one-time experiment in this study. The semi-quantitative nature of the pH indicators, where the assessment of pH is subjective to visual interpretation compared to the standards produced (Figure 1), limits the accuracy when assessing the medium to approximately 0.25 pH units. Values from the 1:3 and 1:1 combinations were combined and graphed over time (Figure 5), using LOESS smoothing with a 95% confidence interval calculated from the standard error of the estimate, with statistical comparisons done in R version 4.1.2 (November 1 2021) [25].

## 4. Conclusions

The combination of bromocresol green and bromocresol purple proved to be a useful addition to standard tissue culture media, allowing for a quick, semi-quantitative visual assessment of the media over time. Varying the ratio of bromocresol green to bromocresol purple alters the pH at which the colour changes from blue to green to yellow, with the 1:3 ratio providing a useful pH range of 5–6.5, while the 1:1 ratio provides a useful pH range of 4.5–6. Both of these pH indicators showed no toxic side effects on the plant species tested in this study and were able to be autoclaved to ensure media sterility. However, additional testing should be done to ensure that there are no potential side effects, as has been reported previously with phenol red acting as a weak oestrogen [5,20,21], although there have been no side effects reported so far for the pH indicators used in this study. The addition of these pH indicators can aid in establishing new species in tissue culture, provide a guide for determining when to change media due to acidification, or add potential buffering agents to limit acidification of the media. We envision using these pH indicators as a ‘traffic light’ system, with blue indicating a high pH, green normal pH, and yellow a low pH in the media, as a quick visual assessment of media pH across the tissue culture collection, useful in some circumstances to quickly screen multiple species (or genotypes within a single species) for their overall effects on the stability of the media pH as a ‘first base’ in optimising tissue culture conditions and media selection.

## Figures and Tables

**Figure 1 plants-12-00740-f001:**
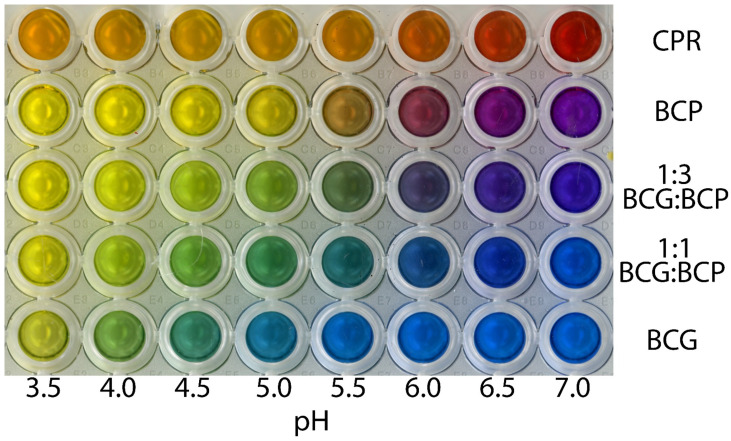
Colour change in chlorophenol red (CPR), bromocresol green (BCG), and bromocresol purple (BCP), as well as a 1:1 and 1:3 combination of BCG:BCP over a pH range of 3.5–7.0. Each well contains 200 µL of the pH indicator at a concentration of 250 µg mL^−1^ in a 0.1 M phosphate buffered solution set to the corresponding pH.

**Figure 2 plants-12-00740-f002:**
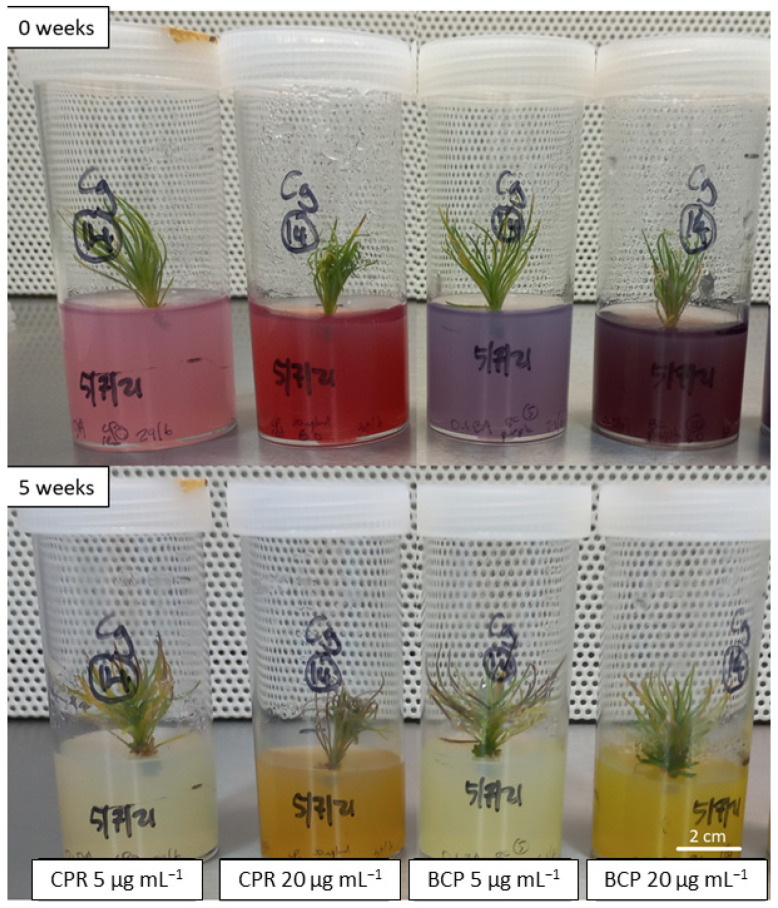
Initial assessment of *Conospermum galeatum* growing on medium containing chlorophenol red (CPR) and bromocresol purple (BCP) at 5 and 20 µg mL^−1^ concentrations for a 5 week period. The initial pH of the medium was approximately 6.0. Colour changes indicate a drop in pH below 5.5 after five weeks, with the colour fading at the 5 µg mL^−1^ concentration.

**Figure 3 plants-12-00740-f003:**
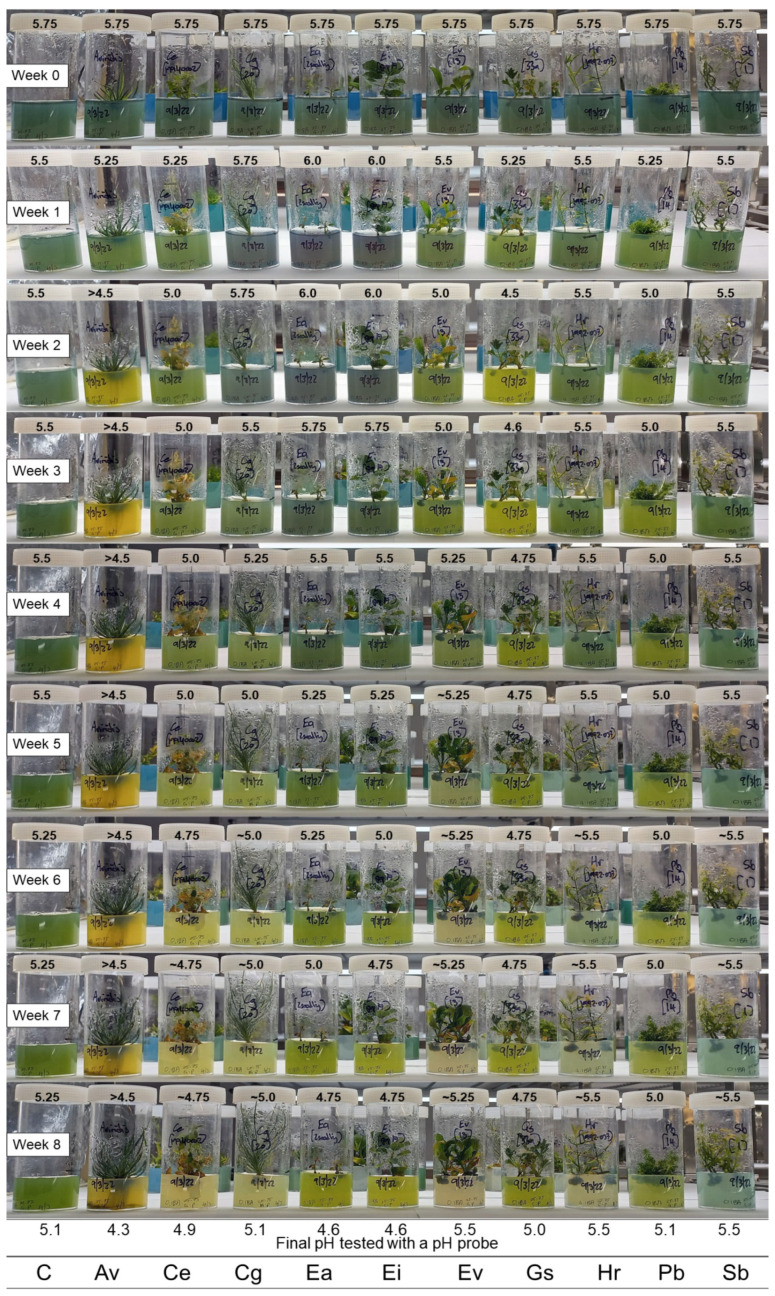
Visual change in pH of the medium with a 1:3 bromocresol green (BCG):bromocresol purple (BCP) combination used at a concentration of 10 µg mL^−1^ over an eight week period. Initial pH of the medium was approximately 5.75, visual estimation of pH based on the colour given above each jar. The symbol ~ denotes pH indicator colour fading in medium, making visual pH estimation difficult. Control (C), *Anigozanthos viridis* (Av), *Commersonia erythrogyna* (Ce), *Conospermum galeatum* (Cg), *Eucalyptus argutifolia* (Ea), *Eucalyptus impensa* (Ei), *Eremophila virens* (Ev), *Grevillea scapigera* (Gs), *Hemiandra rutilans* (Hr), *Philotheca basistyla* (Pb), and *Symonanthus bancroftii* (Sb).

**Figure 4 plants-12-00740-f004:**
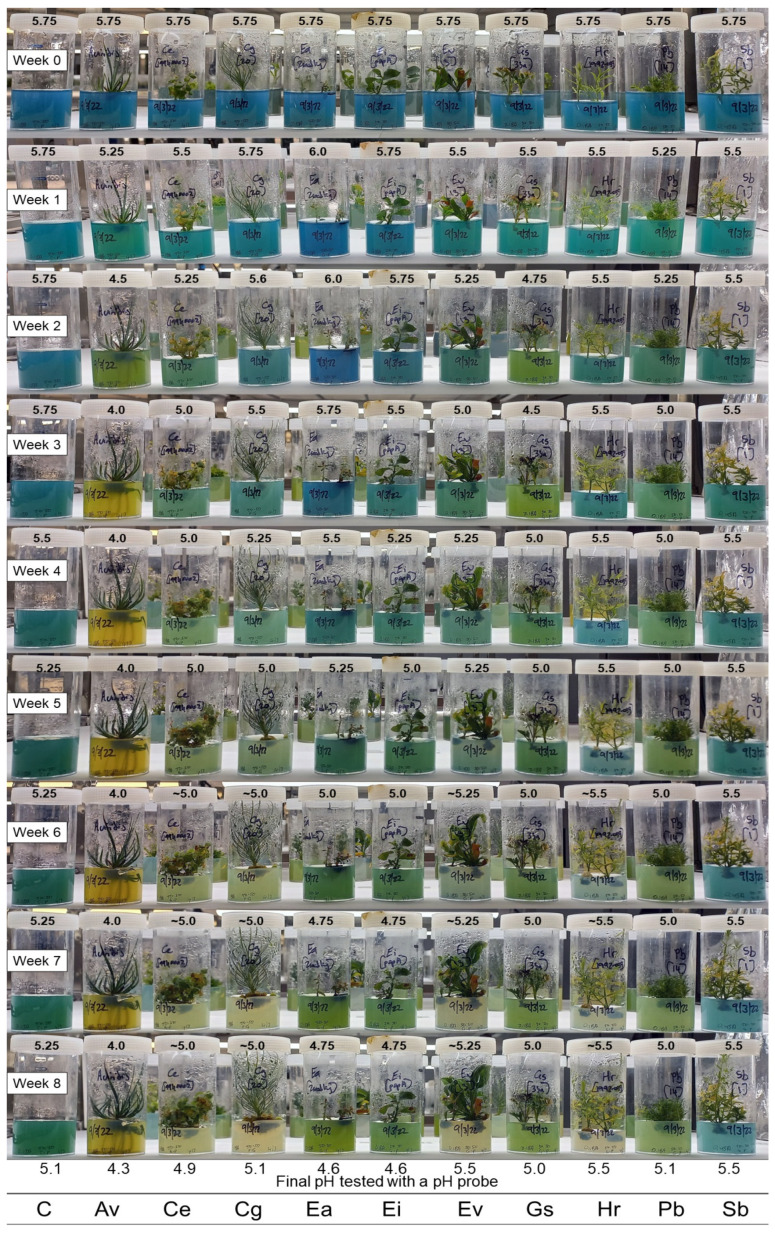
Visual change in pH of the medium with a 1:1 bromocresol green (BCG):bromocresol purple (BCP) combination used at a concentration of 10 µg mL^−1^ over an eight week period. Initial pH of the medium was approximately 5.75, visual estimation of pH based on the colour given above each jar. The symbol ~ denotes pH indicator colour fading in medium, making visual pH estimation difficult. Control (C), *Anigozanthos viridis* (Av), *Commersonia erythrogyna* (Ce), *Conospermum galeatum* (Cg), *Eucalyptus argutifolia* (Ea), *Eucalyptus impensa* (Ei), *Eremophila virens* (Ev)*, Grevillea scapigera* (Gs), *Hemiandra rutilans* (Hr), *Philotheca basistyla* (Pb), and *Symonanthus bancroftii* (Sb).

**Figure 5 plants-12-00740-f005:**
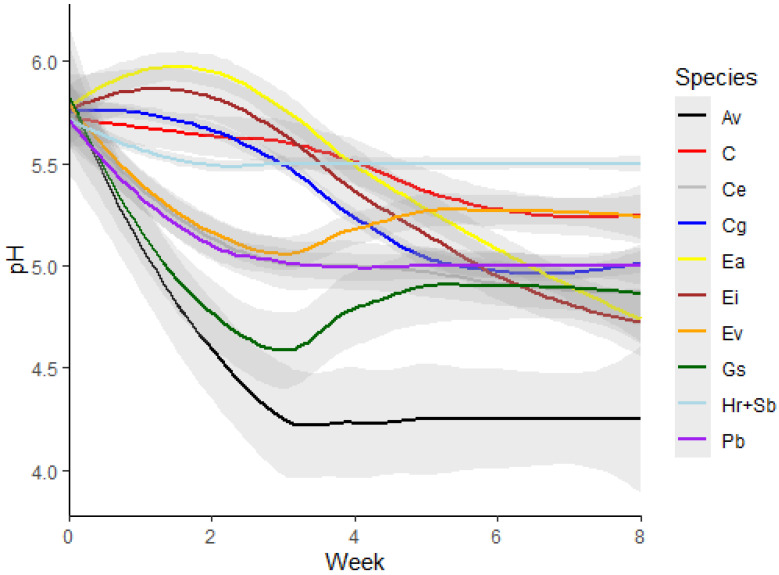
Average pH of the medium (with 95% confidence intervals shown in grey) combining the 1:1 and 1:3 bromocresol green:bromocresol purple combination results used at a concentration of 10 µg mL^−1^ over an eight week period (n = 2). Control (C), *Anigozanthos viridis* (Av), *Commersonia erythrogyna* (Ce), *Conospermum galeatum* (Cg), *Eucalyptus argutifolia* (Ea), *Eucalyptus impensa* (Ei), *Eremophila virens* (Ev)*, Grevillea scapigera* (Gs), *Hemiandra rutilans* and *Symonanthus bancroftii* (Hr+Sb), and *Philotheca basistyla* (Pb). Note: Hr and Sb showed the same pH readings over the eight week period.

## Data Availability

The data are contained within the article.

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
