# Peer review of "A Simple but Effective Combination of pH Indicators for Plant Tissue Culture"

_plants, 2023, doi:10.3390/plants12040740_

Round 1

Reviewer 1 Report

I am rather puzzled by this manuscript which, English-wise, is well written.

While the information provided remains quite technical it still has some interest. However, the reader is expecting some novel insight or an explanation as to why some results have been observed. In absence of these it looks like a preliminary work from a junior student that the authors are trying to publish to support a diploma. In short, we need more!

Some of my main concerns are as follows:

-        It is stated that Conospermum gelatum shoots normally show browning … any reference?

-        Variation range in pH between colour hues was 0.5, and when such “visually estimated” values were compared to “electrode-measured” values there was a 0.25 difference between them … is this not significant?

-        Moreover, no statistics whatsoever was applied???

-        In lines 116-117 you indicate the final pH, around 5.5, was higher than for the control medium … but you said this was 5.7! Either there is a contradiction or there is a mistake here

-        It is also indicated in the text that some cultures were maintained on basal medium for 6-12 months … is this without intermediate subculturing? If so, I would be cautious about any conclusion being drawn from such material. In any case, this must be clarified

-        According to the authors, Eucalypts showed substantial leaf browning by 8 weeks. Was this due to a drop in pH below 5.5 or more likely to explant starvation? Particularly if, as indicated, they were subcultured every 4 weeks. Too many inconsistencies here as with other statements in this manuscript.

-        The section on early detection of bacterial contamination falls like a hair on the soup in the text. It is not supported by any factual evidence or hard data as cast. Therefore, it must be deleted or significantly developed further.

-        Line 183 in M&M states that the initial pH was set at 6.0 but in R&D initial pH was 5.7 and then dropped to 5.5 (see query above). Another contradiction …

Only a very significantly revised version of this manuscript might be reconsidered for publication.

As cast, the manuscript does not meet the minimum requirements of soundness, clarity, statistical significance, novel insight, and proper referencing to be published.

Author Response

We thank the reviewer for their comments and have responded to their concerns raised below:

-        It is stated that Conospermum gelatum shoots normally show browning … any reference?

Unfortunately, no published literature is available for many of the species tested in this manuscript including Conospermum galeatum, these are threatened Australian species that have been initiated into tissue culture for conservation purposes (not for research purposes, so findings are only reported internally in the organisation). Many require additional work to optimise the tissue culture process as little has been done in this area, with this study being the first assessment of media pH for these species. We have added some additional information into the manuscript regarding this point.

-        Variation range in pH between colour hues was 0.5, and when such “visually estimated” values were compared to “electrode-measured” values there was a 0.25 difference between them … is this not significant?

A t-test was done to see if there was a significant difference between the visual estimations and pH probe results. Due to the bigger differences in the visual estimation compared to the pH probe, no significant difference between the visual estimation and probe was seen. The results of the t-test, maximum difference, and standard deviation of the pH probe results vs the visual estimations for each pH combination at week 8 were included in the results and discussion section.

-        Moreover, no statistics whatsoever was applied???

The pH results seen with the two pH combinations show very similar results between the two combinations tested, and some additional results have been included in the manuscript based on these findings. We acknowledge the lack of replication in the article, and this has been clarified in the methods section. However, the results gained from additional replication may provide little additional information.  The visual pH estimations have been re-assessed to more accurately reflect the pH based on the standards when comparing each week as a stand-alone (a better representation of how these pH indicators will practically be used), this limits the accuracy closer to 0.25 pH units, which still provides a useful indication of pH changes in the medium (this information is summarised in the new Figure 5). We have made the point in the manuscript that a more accurate assessment should be done using a pH probe in the species where the pH indicators show large pH changes that may influence shoot growth and health.

-        In lines 116-117 you indicate the final pH, around 5.5, was higher than for the control medium … but you said this was 5.7! Either there is a contradiction or there is a mistake here

This point has been clarified in the text, the control medium (empty jar with no shoots growing in it) showed a slow steady decline in pH over time, these two species final medium pH was higher than the control pH at the end of the 8 weeks (not the starting pH of the control medium).

-        It is also indicated in the text that some cultures were maintained on basal medium for 6-12 months … is this without intermediate subculturing? If so, I would be cautious about any conclusion being drawn from such material. In any case, this must be clarified

We have clarified the point regarding cultures which can be maintained on the same medium for 6-12 months in the manuscript. The study species were chosen for this specific reason, to help understand why some do not require subculturing as frequently as other species - and was this reflected in the pH stability (or otherwise) of the medium. All shoots were placed onto fresh medium at the start of the trial.

-        According to the authors, Eucalypts showed substantial leaf browning by 8 weeks. Was this due to a drop in pH below 5.5 or more likely to explant starvation? Particularly if, as indicated, they were subcultured every 4 weeks. Too many inconsistencies here as with other statements in this manuscript.

The choice of eight weeks over which to conduct this experiment arose from the observation that some of the study species appeared to be unaffected (i.e. did not show obvious signs of stress -such as browning of stems or leaves, abscission or abnormal growth) by extended passages on the same medium (double the normal 4wks or longer) compared to other species which did show signs of stress over much shorter time frames sometimes withing the usual 4 week incubation period. The Eucalyptus species used in this study are subcultured after 4-5wks under standard culture conditions - or their growth and health deteriorates. Unfortunately, we do not currently have more in-depth information on the growth of these species in TC. As time and resources allow, we will proceed with further experiments to optimise culture media for these species now we know media pH may be a confounding factor in their growth. We include them in this study to illustrate how a wide range of species (from various Australian genera and families) presents challenges to media optimisation that can in some part be assisted by simple investigative techniques such as the one described here.

-        The section on early detection of bacterial contamination falls like a hair on the soup in the text. It is not supported by any factual evidence or hard data as cast. Therefore, it must be deleted or significantly developed further.

The section on microbial contamination causing pH changes in the medium has been removed based on the suggestions from reviewer 1 and 2

-        Line 183 in M&M states that the initial pH was set at 6.0 but in R&D initial pH was 5.7 and then dropped to 5.5 (see query above). Another contradiction …

The pH was set to 6 prior to autoclaving. It is well known that the autoclaving process can affect the medium pH and this has been clarified in the text with some references.

Reviewer 2 Report

Bryn Funnekotter and colleagues examined three pH indicators to establish a simple visual method for quickly assessing pH changes in the tissue culture medium. The Authors write in the introduction about the problems with maintaining stable pH conditions of the culture medium. Also, the medium pH level of plant tissue cultures seems essential to many aspects of explant development and growth. Further, the selection of appropriate pH indicators and the conditions for their use (quantity, concentration, sterilization conditions) have their justification, given the importance of in vitro cultures. The introduction seems well-written, and the research problem the Authors set for themselves is sufficiently grounded in the literature and adequately argued, while I have objections to the methodology. Thus, the manuscript requires minor revision. Below is a list of shortcomings that the Authors should consider when preparing the manuscript's revised version.

  1. Although the Authors wrote what pH indicators they tested and at what concentrations and listed the species cultured on MS medium, there needs to be more information regarding how many times the experiment was repeated. Did the Authors base their observations and conclusions on a one-time experiment? I would appreciate clarification and completion of this data in the methodology of the manuscript.
  2. Lines 25-27: It would be good to give examples of specific species that tolerate a broad pH spectrum well, the most commonly used (optimal) and narrow.
  3. Lines 66-67: I would change the order of pH indicators, i.e., instead of bromocresol purple and bromocresol green, I would write bomocresol green and bromocresol purple. Please review the manuscript considering the same order of the pH indicators, as it makes the manuscript easier to be received by the readers. Please have the same order of pH indicators throughout the manuscript.
  4. Figure 5: In image A, no bacterial colonies can be seen, and in image B, the fungal infection is barely visible. There should be a better selection of images for the issue the Authors wanted to describe in subsection 2.3 Early detection of contamination. The ones presented in the manuscript are uninformative.
  5. Line 179: repeated "...with, with...".

In summary, the manuscript is well-written and should be acceptable for publication after addressing the mentioned (minor) issues.

Author Response

We thank the reviewer for their comments on the manuscript, Please see below for our response to their comments.

  1. Although the Authors wrote what pH indicators they tested and at what concentrations and listed the species cultured on MS medium, there needs to be more information regarding how many times the experiment was repeated. Did the Authors base their observations and conclusions on a one-time experiment? I would appreciate clarification and completion of this data in the methodology of the manuscript.

Some additional information has been included in the Methods section on the replication to clarify this point for the readers.

  1. Lines 25-27: It would be good to give examples of specific species that tolerate a broad pH spectrum well, the most commonly used (optimal) and narrow.

Little is known about the pH tolerance of the species listed in this study as this is the first assessment of pH over time done. Further work would need to be done to assess these species tolerance ranges, assessing the media over a wide range of pH starting points, but this is outside the scope of this communication article. Some additional information has been added to the results and discussion section on previous published literature looking at growth at different pH in other species, generally it seems most species in tissue culture have a relatively broad pH tolerance, with growth rates declining once pH drops below 4.5 to 5.5 depending on the species.

  1. Lines 66-67: I would change the order of pH indicators, i.e., instead of bromocresol purple and bromocresol green, I would write bromocresol green and bromocresol purple. Please review the manuscript considering the same order of the pH indicators, as it makes the manuscript easier to be received by the readers. Please have the same order of pH indicators throughout the manuscript.

The order of the pH indicators listed has been changed for consistency with bromocresol green listed before bromocresol purple throughout the article

  1. Figure 5: In image A, no bacterial colonies can be seen, and in image B, the fungal infection is barely visible. There should be a better selection of images for the issue the Authors wanted to describe in subsection 2.3 Early detection of contamination. The ones presented in the manuscript are uninformative.

Figure 5 has been removed in relation to reviewer one’s comments

  1. Line 179: repeated "...with, with...".

Fixed the repeated ‘with’ on line 179

Reviewer 3 Report

The manuscript entitled "A Simple but Effective Combination of pH Indicators for Plant Tissue Culture" was conducted by Funnekotter et al. to examine chlorophenol red, bromocresol purple, and bromocresol green, and their functionality in the growth medium for plant shoot cultures as pH indicators. Also, they assessed a combination of bromocresol green and bromocresol purple for the same reason. They concluded that a combination of bromocresol green and bromocresol purple could be useful as a pH indicator in a standard tissue culture medium to monitor pH over time. The study is on a topic of relevance and general interest to the readers of the journal. I found the manuscript to be overall well prepared. However, I see that it has a poor discussion section. The authors need to provide more discussion of the results and connect them with the previous studies.

Author Response

We thank the reviewer for their comment on our manuscript. 

The discussion has been expanded with additional references, putting our findings into context with previous studies on pH indicators as suggested by reviewers 1 and 2. However as previously stated in our comments to reviewer 1, no published literature is available for many of the species tested in this manuscript as these are threatened Australian species that have been initiated into tissue culture for conservation purposes (not for research purposes, so findings are only reported internally within the organisation). Little is known about the pH tolerance of the species listed in this study as this is the first assessment of pH over time done.

Round 2

Reviewer 1 Report

I am accepting you submission even if I remain rather sceptical.

My main concern is that you state yourselves that you still do not know which are the best conditions for growth in vitro of many of these species ... yet, you say that there are no side effects of the addition of the indicators and that they moreover help to optimise the composition of the media to be used. I still see a contradiction between these statements.

I must admit I would have felt much more at ease had you included at least one species for which you did know the optimum conditions and where you would have only have changed the medium pH to assess the effects of this on growth.